# Is the Bellman residual a bad proxy?

**Matthieu Geist**[1], **Bilal Piot**[2,3] **and Olivier Pietquin** [2,3]
[1] Université de Lorraine & CNRS, LIEC, UMR 7360, Metz, F-57070 France
[2] Univ. Lille, CNRS, Centrale Lille, Inria, UMR 9189 - CRIStAL, F-59000 Lille, France
[3] Now with Google DeepMind, London, United Kingdom
matthieu.geist@univ-lorraine.fr
bilal.piot@univ-lille1.fr, olivier.pietquin@univ-lille1.fr

## Abstract

This paper aims at theoretically and empirically comparing two standard optimization criteria for Reinforcement Learning: i) maximization of the mean value and ii) minimization of the Bellman residual. For that purpose, we place ourselves in the framework of policy search algorithms, that are usually designed to maximize the mean value, and derive a method that minimizes the residual $\|T_* v_\pi - v_\pi\|_{1,\nu}$ over policies. A theoretical analysis shows how good this proxy is to policy optimization, and notably that it is better than its value-based counterpart. We also propose experiments on randomly generated generic Markov decision processes, specifically designed for studying the influence of the involved concentrability coefficient. They show that the Bellman residual is generally a bad proxy to policy optimization and that directly maximizing the mean value is much better, despite the current lack of deep theoretical analysis. This might seem obvious, as directly addressing the problem of interest is usually better, but given the prevalence of (projected) Bellman residual minimization in value-based reinforcement learning, we believe that this question is worth to be considered.

## 1   Introduction

Reinforcement Learning (RL) aims at estimating a policy $\pi$ close to the optimal one, in the sense that its value, $v_\pi$ (the expected discounted return), is close to maximal, *i.e* $\|v_* - v_\pi\|$ is small ($v_*$ being the optimal value), for some norm. Controlling the residual $\|T_* v_\theta - v_\theta\|$ (where $T_*$ is the optimal Bellman operator and $v_\theta$ a value function parameterized by $\theta$) over a class of parameterized value functions is a classical approach in value-based RL, and especially in Approximate Dynamic Programming (ADP). Indeed, controlling this residual allows controlling the distance to the optimal value function: generally speaking, we have that

$$\|v_* - v_{\pi_{v_\theta}}\| \leq \frac{C}{1 - \gamma} \|T_* v_\theta - v_\theta\|, \tag{1}$$

with the policy $\pi_{v_\theta}$ being greedy with respect to $v_\theta$ [17, 19].

Some classical ADP approaches actually minimize a projected Bellman residual, $\|\Pi(T_* v_\theta - v_\theta)\|$, where $\Pi$ is the operator projecting onto the hypothesis space to which $v_\theta$ belongs: Approximate Value Iteration (AVI) [11, 9] tries to minimize this using a fixed-point approach, $v_{\theta_{k+1}} = \Pi T_* v_{\theta_k}$, and it has been shown recently [18] that Least-Squares Policy Iteration (LSPI) [13] tries to minimize it using a Newton approach[1]. Notice that in this case (projected residual), there is no general performance bound[2] for controlling $\|v_* - v_{\pi_{v_\theta}}\|$.

Despite the fact that (unprojected) residual approaches come easily with performance guarantees, they are not extensively studied in the (value-based) literature (one can mention [3] that considers a subgradient descent or [19] that frames the norm of the residual as a delta-convex function). A reason for this is that they lead to biased estimates when the Markovian transition kernel is stochastic and unknown [1], which is a rather standard case. Projected Bellman residual approaches are more common, even if not introduced as such originally (notable exceptions are [16, 18]).

An alternative approach consists in maximizing directly the mean value $\mathbb{E}_\nu[v_\pi(S)]$ for a user-defined state distribution $\nu$, this being equivalent to directly minimizing $\|v_* - v_\pi\|_{1,\nu}$, see Sec. 2. This suggests defining a class of parameterized policies and optimizing over them, which is the predominant approach in policy search[3] [7].

This paper aims at theoretically and experimentally studying these two approaches: maximizing the mean value (related algorithms operate on policies) and minimizing the residual (related algorithms operate on value functions). In that purpose, we place ourselves in the context of policy search algorithms. We adopt this position because we could derive a method that minimizes the residual $\|T_* v_\pi - v_\pi\|_{1,\nu}$ over policies and compare to other methods that usually maximize the mean value. On the other hand, adapting ADP methods so that they maximize the mean value is way harder[4]. This new approach is presented in Sec. 3, and we show theoretically how good this proxy.

In Sec. 4, we conduct experiments on randomly generated generic Markov decision processes to compare both approaches empirically. The experiments are specifically designed to study the influence of the involved concentrability coefficient. Despite the good theoretical properties of the Bellman residual approach, it turns out that it only works well if there is a good match between the sampling distribution and the discounted state occupancy distribution induced by the optimal policy, which is a very limiting requirement. In comparison, maximizing the mean value is rather insensitive to this issue and works well whatever the sampling distribution is, contrary to what suggests the sole related theoretical bound. This study thus suggests that maximizing the mean value, although it doesn't provide easy theoretical analysis, is a better approach to build efficient and robust RL algorithms.

## 2 Background

### 2.1 Notations

Let $\Delta_X$ be the set of probability distributions over a finite set $X$ and $Y^X$ the set of applications from $X$ to the set $Y$. By convention, all vectors are column vectors, except distributions (for left multiplication). A Markov Decision Process (MDP) is a tuple $\{\mathcal{S}, \mathcal{A}, P, \mathcal{R}, \gamma\}$, where $\mathcal{S}$ is the finite state space[5], $\mathcal{A}$ is the finite action space, $P \in (\Delta_\mathcal{S})^{\mathcal{S} \times \mathcal{A}}$ is the Markovian transition kernel ($P(s'|s,a)$ denotes the probability of transiting to $s'$ when action $a$ is applied in state $s$), $\mathcal{R} \in \mathbb{R}^{\mathcal{S} \times \mathcal{A}}$ is the bounded reward function ($\mathcal{R}(s,a)$ represents the local benefit of doing action $a$ in state $s$) and $\gamma \in (0,1)$ is the discount factor. For $v \in \mathbb{R}^\mathcal{S}$, we write $\|v\|_{1,\nu} = \sum_{s \in \mathcal{S}} \nu(s)|v(s)|$ the $\nu$-weighted $\ell_1$-norm of $v$.

Notice that when the function $v \in \mathbb{R}^\mathcal{S}$ is componentwise positive, that is $v \geq 0$, the $\nu$-weighted $\ell_1$-norm of $v$ is actually its expectation with respect to $\nu$: if $v \geq 0$, then $\|v\|_{1,\nu} = \mathbb{E}_\nu[v(S)] = \nu v$. We will make an intensive use of this basic property in the following.

A stochastic policy $\pi \in (\Delta_\mathcal{A})^\mathcal{S}$ associates a distribution over actions to each state. The policy-induced reward and transition kernels, $\mathcal{R}_\pi \in \mathbb{R}^\mathcal{S}$ and $P_\pi \in (\Delta_\mathcal{S})^\mathcal{S}$, are defined as

$$\mathcal{R}_\pi(s) = \mathbb{E}_{\pi(.|s)}[\mathcal{R}(s,A)] \text{ and } P_\pi(s'|s) = \mathbb{E}_{\pi(.|s)}[P(s'|s,A)].$$

The quality of a policy is quantified by the associated value function $v_\pi \in \mathbb{R}^\mathcal{S}$:

$$v_\pi(s) = \mathbb{E}[\sum_{t \geq 0} \gamma^t \mathcal{R}_\pi(S_t)|S_0 = s, S_{t+1} \sim P_\pi(.|S_t)].$$

The value $v_\pi$ is the unique fixed point of the Bellman operator $T_\pi$, defined as $T_\pi v = \mathcal{R}_\pi + \gamma P_\pi v$ for any $v \in \mathbb{R}^{\mathcal{S}}$. Let define the second Bellman operator $T_*$ as, for any $v \in \mathbb{R}^{\mathcal{S}}$, $T_* v = \max_{\pi \in (\Delta_{\mathcal{A}})^{\mathcal{S}}} T_\pi v$. A policy $\pi$ is greedy with respect to $v \in \mathbb{R}^{\mathcal{S}}$, denoted $\pi \in \mathcal{G}(v)$ if $T_\pi v = T_* v$. There exists an optimal policy $\pi_*$ that satisfies componentwise $v_{\pi_*} \geq v_\pi$, for all $\pi \in (\Delta_{\mathcal{A}})^{\mathcal{S}}$. Moreover, we have that $\pi_* \in \mathcal{G}(v_*)$, with $v_*$ being the unique fixed point of $T_*$.

Finally, for any distribution $\mu \in \Delta_{\mathcal{S}}$, the $\gamma$-weighted occupancy measure induced by the policy $\pi$ when the initial state is sampled from $\mu$ is defined as

$$d_{\mu,\pi} = (1 - \gamma)\mu \sum_{t \geq 0} \gamma^t P_\pi^t = (1 - \gamma)\mu(I - \gamma P_\pi)^{-1} \in \Delta_{\mathcal{S}}.$$

For two distributions $\mu$ and $\nu$, we write $\left\| \frac{\mu}{\nu} \right\|_\infty$ the smallest constant $C$ satisfying, for all $s \in \mathcal{S}$, $\mu(s) \leq C\nu(s)$. This quantity measures the mismatch between the two distributions.

## 2.2 Maximizing the mean value

Let $\mathcal{P}$ be a space of parameterized stochastic policies and let $\mu$ be a distribution of interest. The optimal policy has a higher value than any other policy, for *any* state. If the MDP is too large, satisfying this condition is not reasonable. Therefore, a natural idea consists in searching for a policy such that the associated value function is as close as possible to the optimal one, *in expectation*, according to a distribution of interest $\mu$. More formally, this means minimizing $\|v_* - v\|_{1,\mu} = \mathbb{E}_\mu[v_*(S) - v_\pi(S)] \geq 0$. The optimal value function being unknown, one cannot address this problem directly, but it is equivalent to maximizing $\mathbb{E}_\mu[v_\pi(S)]$.

This is the basic principle of many policy search approaches:

$$\max_{\pi \in \mathcal{P}} J_\nu(\pi) \text{ with } J_\nu(\pi) = \mathbb{E}_\nu[v_\pi(S)] = \nu v_\pi.$$

Notice that we used a *sampling* distribution $\nu$ here, possibly different from the distribution *of interest* $\mu$. Related algorithms differ notably by the considered criterion (*e.g.*, it can be the mean reward rather than the $\gamma$-discounted cumulative reward considered here) and by how the corresponding optimization problem is solved. We refer to [7] for a survey on that topic.

Contrary to ADP, the theoretical efficiency of this family of approaches has not been studied a lot. Indeed, as far as we know, there is a sole performance bound for maximizing the mean value.

**Theorem 1** (Scherrer and Geist [22]). *Assume that the policy space $\mathcal{P}$ is stable by stochastic mixture, that is $\forall \pi, \pi' \in \mathcal{P}, \forall \alpha \in (0, 1), \quad (1 - \alpha)\pi + \alpha\pi' \in \mathcal{P}$. Define the $\nu$-greedy-complexity of the policy space $\mathcal{P}$ as*

$$\mathcal{E}_\nu(\mathcal{P}) = \max_{\pi \in \mathcal{P}} \min_{\pi' \in \mathcal{P}} d_{\nu,\pi}(T_* v_\pi - T_{\pi'} v_\pi).$$

*Then, any policy $\pi$ that is an $\epsilon$-local optimum of $J_\nu$, in the sense that*

$$\forall \pi' \in \Pi, \quad \lim_{\alpha \to 0} \frac{\nu v_{(1-\alpha)\pi + \alpha\pi'} - \nu v_\pi}{\alpha} \leq \epsilon,$$

*enjoys the following global performance guarantee:*

$$\mu(v_* - v_\pi) \leq \frac{1}{(1 - \gamma)^2} \left\| \frac{d_{\mu,\pi*}}{\nu} \right\|_\infty (\mathcal{E}_\nu(\mathcal{P}) + \epsilon).$$

This bound (as all bounds of this kind) has three terms: an horizon term, a concentrability term and an error term. The term $\frac{1}{1-\gamma}$ is the average optimization horizon. This concentrability coefficient $(\|d_{\mu,\pi*}/\nu\|_\infty)$ measures the mismatch between the used distribution $\nu$ and the $\gamma$-weighted occupancy measure induced by the *optimal* policy $\pi_*$ when the initial state is sampled from the distribution of interest $\mu$. This tells that if $\mu$ is the distribution of interest, one should optimize $J_{d_{\mu,\pi_*}}$, which is not feasible, $\pi_*$ being unknown (in this case, the coefficient is equal to 1, its lower bound). This coefficient can be arbitrarily large: consider the case where $\mu$ concentrates on a single starting state (that is $\mu(s_0) = 1$ for a given state $s_0$) and such that the optimal policy leads to other states (that is, $d_{\mu,\pi_*}(s_0) < 1$), the coefficient is then infinite. However, it is also the best concentrability coefficient according to [21], that provides a theoretical and empirical comparison of Approximate Policy Iteration (API) schemes. The error term is $\mathcal{E}_\nu(\mathcal{P}) + \epsilon$, where $\mathcal{E}_\nu(\mathcal{P})$ measures the capacity of

the policy space to represent the policies being greedy with respect to the value of any policy in $\mathcal{P}$ and $\epsilon$ tells how the computed policy $\pi$ is close to a local optimum of $J_\nu$.

There exist other policy search approches, based on ADP rather than on maximizing the mean value, such as Conservative Policy Iteration (CPI) [12] or Direct Policy Iteration (DPI) [14]. The bound of Thm. 1 matches the bounds of DPI or CPI. Actually, CPI can be shown to be a boosting approach maximizing the mean value. See the discussion in [22] for more details. However, this bound is also based on a very strong assumption (stability by stochastic mixture of the policy space) which is not satisfied by all commonly used policy parameterizations.

## 3   Minimizing the Bellman residual

Direct maximization of the mean value operates on policies, while residual approaches operate on value functions. To study these two optimization criteria together, we introduce a policy search method that minimizes a residual. As noted before, we do so because it is much simpler than introducing a value-based approach that maximizes the mean value. We also show how good this proxy is to policy optimization. Although this algorithm is new, it is not claimed to be a core contribution of the paper. Yet it is clearly a mandatory step to support the comparison between optimization criteria.

### 3.1   Optimization problem

We propose to search a policy in $\mathcal{P}$ that minimizes the following Bellman residual:

$$\min_{\pi \in \mathcal{P}} \mathcal{J}_\nu(\pi) \text{ with } \mathcal{J}_\nu(\pi) = \|T_* v_\pi - v_\pi\|_{1,\nu}.$$

Notice that, as for the maximization of the mean value, we used a *sampling* distribution $\nu$, possibly different from the distribution *of interest* $\mu$.

From the basic properties of the Bellman operator, for any policy $\pi$ we have that $T_* v_\pi \geq v_\pi$. Consequently, the $\nu$-weighted $\ell_1$-norm of the residual is indeed the *expected* Bellman residual:

$$\mathcal{J}_\nu(\pi) = \mathbb{E}_\nu[[T_* v_\pi](S) - v_\pi(S)] = \nu(T_* v_\pi - v_\pi).$$

Therefore, there is naturally no bias problem for minimizing a residual here, contrary to other residual approaches [1]. This is an interesting result on its own, as removing the bias in value-based residual approaches is far from being straightforward. This results from the optimization being done over policies and not over values, and thus from $v_\pi$ being an actual value (the one of the current policy) obeying to the Bellman equation[6].

Any optimization method can be envisioned to minimize $\mathcal{J}_\nu$. Here, we simply propose to apply a subgradient descent (despite the lack of convexity).

**Theorem 2** (Subgradient of $\mathcal{J}_\nu$). *Recall that given the considered notations, the distribution $\nu P_{\mathcal{G}(v_\pi)}$ is the state distribution obtained by sampling the initial state according to $\nu$, applying the action being greedy with respect to $v_\pi$ and following the dynamics to the next state. This being said, the subgradient of $\mathcal{J}_\nu$ is given by*

$$-\nabla \mathcal{J}_\nu(\pi) = \frac{1}{1-\gamma} \sum_{s,a} \left( d_{\nu,\pi}(s) - \gamma d_{\nu P_{\mathcal{G}(v_\pi)},\pi}(s) \right) \pi(a|s) \nabla \ln \pi(a|s) q_\pi(s,a),$$

*with $q_\pi(s,a) = \mathcal{R}(s,a) + \gamma \sum_{s' \in \mathcal{S}} P(s'|s,a) v_\pi(s')$ the state-action value function.*

*Proof.* The proof relies on basic (sub)gradient calculus, it is given in the appendix. □

There are two terms in the negative subgradient $-\nabla \mathcal{J}_\nu$: the first one corresponds to the gradient of $J_\nu$, the second one (up to the multiplication by $-\gamma$) is the gradient of $J_{\nu P_{\mathcal{G}(v_\pi)}}$ and acts as a kind of correction. This subgradient can be estimated using Monte Carlo rollouts, but doing so is harder than for classic policy search (as it requires additionally sampling from $\nu P_{\mathcal{G}(v_\pi)}$, which requires estimating

the state-action value function). Also, this gradient involves computing the maximum over actions (as it requires sampling from $\nu P_{\mathcal{G}(v_\pi)}$, that comes from explicitly considering the Bellman optimality operator), which prevents from extending easily this approach to continuous actions, contrary to classic policy search.

Thus, from an algorithmic point of view, this approach has drawbacks. Yet, we do not discuss further how to efficiently estimate this subgradient since we introduced this approach for the sake of comparison to standard policy search methods only. For this reason, we will consider an ideal algorithm in the experimental section where an analytical computation of the subgradient is possible, see Sec. 4. This will place us in an unrealistically good setting, which will help focusing on the main conclusions. Before this, we study how good this proxy is to policy optimization.

## 3.2 Analysis

**Theorem 3** (Proxy bound for residual policy search). *We have that*

$$\|v_* - v_\pi\|_{1,\mu} \leq \frac{1}{1-\gamma} \left\| \frac{d_{\mu,\pi_*}}{\nu} \right\|_\infty \mathcal{J}_\nu(\pi) = \frac{1}{1-\gamma} \left\| \frac{d_{\mu,\pi_*}}{\nu} \right\|_\infty \|T_* v_\pi - v_\pi\|_{1,\nu}.$$

*Proof.* The proof can be easily derived from the analyses of [12], [17] or [22]. We detail it for completeness in the appendix. □

This bound shows how controlling the residual helps in controlling the error. It has a linear dependency on the horizon and the concentrability coefficient is the best one can expect (according to [21]). It has the same form has the bounds for value-based residual minimization [17, 19] (see also Eq. (1)). It is even better due to the involved concentrability coefficient (the ones for value-based bounds are worst, see [21] for a comparison).

Unfortunately, this bound is hardly comparable to the one of Th. 1, due to the error terms. In Th. 3, the error term (the residual) is a global error (how good is the residual as a proxy), whereas in Th. 1 the error term is mainly a local error (how small is the gradient after minimizing the mean value). Notice also that Th. 3 is roughly an intermediate step for proving Th. 1, and that it applies to any policy (suggesting that searching for a policy that minimizes the residual makes sense). One could argue that a similar bound for mean value maximization would be something like: if $J_\mu(\pi) \geq \alpha$, then $\|v_* - v_\pi\|_{1,\mu} \leq \mu v_* - \alpha$. However, this is an oracle bound, as it depends on the unknown solution $v_*$. It is thus hardly exploitable.

The aim of this paper is to compare these two optimization approaches to RL. At a first sight, maximizing directly the mean value should be better (as a more direct approach). If the bounds of Th. 1 and 3 are hardly comparable, we can still discuss the involved terms. The horizon term is better (linear instead of quadratic) for the residual approach. Yet, an horizon term can possibly be hidden in the residual itself. Both bounds imply the same concentrability coefficient, the best one can expect. This is a very important term in RL bounds, often underestimated: as these coefficients can easily explode, minimizing an error makes sense only if it's not multiplied by infinity. This coefficient suggests that one should use $d_{\mu,\pi_*}$ as the sampling distribution. This is rarely reasonable, while using instead directly the distribution of interest is more natural. Therefore, the experiments we propose on the next section focus on the influence of this concentrability coefficient.

## 4 Experiments

We consider Garnet problems [2, 4]. They are a class of randomly built MDPs meant to be totally abstract while remaining representative of the problems that might be encountered in practice. Here, a Garnet $G(|\mathcal{S}|, |\mathcal{A}|, b)$ is specified by the number of states, the number of actions and the branching factor. For each $(s, a)$ couple, $b$ different next states are chosen randomly and the associated probabilities are set by randomly partitioning the unit interval. The reward is null, except for $10\%$ of states where it is set to a random value, uniform in $(1, 2)$. We set $\gamma = 0.99$.

For the policy space, we consider a Gibbs parameterization: $\mathcal{P} = \{\pi_w : \pi_w(a|s) \propto e^{w^\top \phi(s,a)}\}$. The features are also randomly generated, $F(d, l)$. First, we generate binary state-features $\varphi(s)$ of dimension $d$, such that $l$ components are set to 1 (the others are thus 0). The positions of the 1's are

selected randomly such that no two states have the same feature. Then, the state-action features, of dimension $d|\mathcal{A}|$, are classically defined as $\phi(s,a) = (0 \ldots 0 \quad \varphi(s) \quad 0 \ldots 0)^\top$, the position of the zeros depending on the action. Notice that in general this policy space is not stable by stochastic mixture, so the bound for policy search does not formally apply.

We compare classic policy search (denoted as PS($\nu$)), that maximizes the mean value, and residual policy search (denoted as RPS($\nu$)), that minimizes the mean residual. We optimize the relative objective functions with a normalized gradient ascent (resp. normalized subgradient descent) with a constant learning rate $\alpha = 0.1$. The gradients are computed analytically (as we have access to the model), so the following results represent an ideal case, when one can do an infinite number of rollouts. Unless said otherwise, the distribution $\mu \in \Delta_\mathcal{S}$ of interest is the uniform distribution.

## 4.1 Using the distribution of interest

First, we consider $\nu = \mu$. We generate randomly 100 Garnets $G(30, 4, 2)$ and 100 features $F(8, 3)$. For each Garnet-feature couple, we run both algorithms for $T = 1000$ iterations. For each algorithm, we measure two quantities: the (normalized) error $\frac{\|v_* - v_\pi\|_{1,\mu}}{\|v_*\|_{1,\mu}}$ (notice that as rewards are positive, we have $\|v_*\|_{1,\mu} = \mu v_*$) and the Bellman residual $\|T_* v_\pi - v_\pi\|_{1,\mu}$, where $\pi$ depends on the algorithm and on the iteration. We show the results (mean$\pm$standard deviation) on Fig. 1.

a. Error for PS($\mu$).　　b. Error for RPS($\mu$).　　c. Residual for PS($\mu$).　　d. Residual for RPS($\mu$).

Figure 1: Results on the Garnet problems, when $\nu = \mu$.

Fig. 1.a shows that PS($\mu$) succeeds in decreasing the error. This was to be expected, as it is the criterion it optimizes. Fig. 1.c shows how the residual of the policies computed by PS($\mu$) evolves. By comparing this to Fig. 1.a, it can be observed that the residual and the error are not necessarily correlated: the error can decrease while the residual increases, and a low error does not necessarily involves a low residual.

Fig. 1.d shows that RPS($\mu$) succeeds in decreasing the residual. Again, this is not surprising, as it is the optimized criterion. Fig. 1.b shows how the error of the policies computed by RPS($\mu$) evolves. Comparing this to Fig. 1.d, it can be observed that decreasing the residual lowers the error: this is consistent with the bound of Thm. 3.

Comparing Figs. 1.a and 1.b, it appears clearly that RPS($\mu$) is less efficient than PS($\mu$) for decreasing the error. This might seem obvious, as PS($\mu$) directly optimizes the criterion of interest. However, when comparing the errors and the residuals for each method, it can be observed that they are not necessarily correlated. Decreasing the residual lowers the error, but one can have a low error with a high residual and vice versa.

As explained in Sec. 1, (projected) residual-based methods are prevalent for many reinforcement learning approaches. We consider a policy-based residual rather than a value-based one to ease the comparison, but it is worth studying the reason for such a different behavior.

## 4.2 Using the ideal distribution

The lower the concentrability coefficient $\|\frac{d_{\mu,\pi_*}}{\nu}\|_\infty$ is, the better the bounds in Thm. 1 and 3 are. This coefficient is minimized for $\nu = d_{\mu,\pi_*}$. This is an unrealistic case ($\pi_*$ is unknown), but since we work with known MDPs we can compute this quantity (the model being known), for the sake of a complete empirical analysis. Therefore, PS($d_{\mu,\pi_*}$) and RPS($d_{\mu,\pi_*}$) are compared in Fig. 2. We highlight the fact that the errors and the residuals shown in this figure are measured respectively to the distribution of interest $\mu$, and not the distribution $d_{\mu,\pi_*}$ used for the optimization.

a. Error for PS($d_{\mu,\pi_*}$). b. Error for RPS($d_{\mu,\pi_*}$).     c. Residual for        d. Residual for

                                                                PS($d_{\mu,\pi_*}$).        RPS($d_{\mu,\pi_*}$).

Figure 2: Results on the Garnet problems, when $\nu = d_{\mu,\pi_*}$.

Fig. 2.a shows that PS($d_{\mu,\pi_*}$) succeeds in decreasing the error $\|v_* - v_\pi\|_{1,\mu}$. However, comparing Fig. 2.a to Fig. 1.a, there is no significant gain in using $\nu = d_{\mu,\pi_*}$ instead of $\nu = \mu$. This suggests that the dependency of the bound in Thm. 1 on the concentrability coefficient is not tight. Fig. 2.c shows how the corresponding residual evolves. Again, there is no strong correlation between the residual and the error.

Fig. 2.d shows how the residual $\|T_*v_\pi - v_\pi\|_{1,\mu}$ evolves for RPS($d_{\mu,\pi_*}$). It is not decreasing, but it is not what is optimized (the residual $\|T_*v_\pi - v_\pi\|_{1,d_{\mu,\pi_*}}$, not shown, decreases indeed, in a similar fashion than Fig. 1.d). Fig. 2.b shows how the related error evolves. Compared to Fig. 2.a, there is no significant difference. The behavior of the residual is similar for both methods (Figs. 2.c and 2.d).

Overall, this suggests that controlling the residual (RPS) allows controlling the error, but that this requires a wise choice for the distribution $\nu$. On the other hand, controlling directly the error (PS) is much less sensitive to this. In other words, this suggests a stronger dependency of the residual approach to the mismatch between the sampling distribution and the discounted state occupancy measure induced by the optimal policy.

## 4.3 Varying the sampling distribution

This experiment is designed to study the effect of the mismatch between the distributions. We sample 100 Garnets $G(30, 4, 2)$, as well as associated feature sets $F(8, 3)$. The distribution of interest is no longer the uniform distribution, but a measure that concentrates on a single starting state of interest $s_0$: $\mu(s_0) = 1$. This is an adverserial case, as it implies that $\|\frac{d_{\mu,\pi_*}}{\mu}\|_\infty = \infty$: the branching factor being equal to 2, the optimal policy $\pi_*$ cannot concentrate on $s_0$.

The sampling distribution is defined as being a mixture between the distribution of interest and the ideal distribution. For $\alpha \in [0, 1]$, $\nu_\alpha$ is defined as $\nu_\alpha = (1 - \alpha)\mu + \alpha d_{\mu,\pi_*}$. It is straightforward to show that in this case the concentrability coefficient is indeed $\frac{1}{\alpha}$ (with the convention that $\frac{1}{0} = \infty$):

$$\left\|\frac{d_{\mu,\pi_*}}{\nu_\alpha}\right\|_\infty = \max\left(\frac{d_{\mu,\pi_*}(s_0)}{(1 - \alpha) + \alpha d_{\mu,\pi_*}(s_0)}; \frac{1}{\alpha}\right) = \frac{1}{\alpha}.$$

For each MDP, the learning (for PS($\nu_\alpha$) and RPS($\nu_\alpha$)) is repeated, from the same initial policy, by setting $\alpha = \frac{1}{k}$, for $k \in [1; 25]$. Let $\pi_{t,\mathrm{x}}$ be the policy learnt by algorithm x (PS or RPS) at iteration $t$, the integrated error (resp. integrated residual) is defined as

$$\frac{1}{T}\sum_{t=1}^{T}\frac{\|v_* - v_{\pi_{t,\mathrm{x}}}\|_{1,\mu}}{\|v_*\|_{1,\mu}} \quad \left(\text{resp. } \frac{1}{T}\sum_{t=1}^{T}\|T_*v_{\pi_{t,\mathrm{x}}} - v_{\pi_{t,\mathrm{x}}}\|_{1,\mu}\right).$$

Notice that here again, the integrated error and residual are defined with respect to $\mu$, the distribution of interest, and not $\nu_\alpha$, the sampling distribution used for optimization. We get an integrated error (resp. residual) for each value of $\alpha = \frac{1}{k}$, and represent it as a function of $k = \|\frac{d_{\mu,\pi_*}}{\nu_\alpha}\|_\infty$, the concentrability coefficient. Results are presented in Fig. 3, that shows these functions averaged across the 100 randomly generated MDPs (mean±standard deviation as before, minimum and maximum values are shown in dashed line).

Fig. 3.a shows the integrated error for PS($\nu_\alpha$). It can be observed that the mismatch between measures has no influence on the efficiency of the algorithm. Fig. 3.b shows the same thing for RPS($\nu_\alpha$). The integrated error increases greatly as the mismatch between the sampling measure and the ideal one

| a. Integrated error for PS($\nu_\alpha$). | b. Integrated error for RPS($\nu_\alpha$). | c. Integrated residual for PS($\nu_\alpha$). | d. Integrated residual for RPS($\nu_\alpha$). |

Figure 3: Results for the sampling distribution $\nu_\alpha$.

increases (the value to which the error saturates correspond to no improvement over the initial policy). Comparing both figures, it can be observed that RPS performs as well as PS only when the ideal distribution is used (this corresponds to a concentrability coefficient of 1). Fig. 3.c and 3.d show the integrated residual for each algorithm. It can be observed that RPS consistently achieves a lower residual than PS.

Overall, this suggests that using the Bellman residual as a proxy is efficient only if the sampling distribution is close to the ideal one, which is difficult to achieve in general (the ideal distribution $d_{\mu,\pi_*}$ being unknown). On the other hand, the more direct approach consisting in maximizing the mean value is much more robust to this issue (and can, as a consequence, be considered directly with the distribution of interest).

One could argue that the way we optimize the considered objective function is rather naive (for example, considering a constant learning rate). But this does not change the conclusions of this experimental study, that deals with how the error and the Bellman residual are related and with how the concentrability influences each optimization approach. This point is developed in the appendix.

## 5   Conclusion

The aim of this article was to compare two optimization approaches to reinforcement learning: minimizing a Bellman residual and maximizing the mean value. As said in Sec. 1, Bellman residuals are prevalent in ADP. Notably, value iteration minimizes such a residual using a fixed-point approach and policy iteration minimizes it with a Newton descent. On another hand, maximizing the mean value (Sec. 2) is prevalent in policy search approaches.

As Bellman residual minimization methods are naturally value-based and mean value maximization approaches policy-based, we introduced a policy-based residual minimization algorithm in order to study both optimization problems together. For the introduced residual method, we proved a proxy bound, better than value-based residual minimization. The different nature of the bounds of Th. 1 and 3 made the comparison difficult, but both involve the same concentrability coefficient, a term often underestimated in RL bounds.

Therefore, we compared both approaches empirically on a set of randomly generated Garnets, the study being designed to quantify the influence of this concentrability coefficient. From these experiments, it appears that the Bellman residual is a good proxy for the error (the distance to the optimal value function) only if, luckily, the concentrability coefficient is small for the considered MDP and the distribution of interest, or one can afford a change of measure for the optimization problem, such that the sampling distribution is close to the ideal one. Regarding this second point, one can change to a measure different from the ideal one, $d_{\mu,\pi_*}$ (for example, using for $\nu$ a uniform distribution when the distribution of interest concentrates on a single state would help), but this is difficult in general (one should know roughly where the optimal policy will lead to). Conversely, maximizing the mean value appears to be insensitive to this problem. This suggests that the Bellman residual is generally a bad proxy to policy optimization, and that maximizing the mean value is more likely to result in efficient and robust reinforcement learning algorithms, despite the current lack of deep theoretical analysis.

This conclusion might seems obvious, as maximizing the mean value is a more direct approach, but this discussion has never been addressed in the literature, as far as we know, and we think it to be important, given the prevalence of (projected) residual minimization in value-based RL.

## Footnotes

[1](Exact) policy iteration actually minimizes $\|T_* v - v\|$ using a Newton descent [10].

[2]With a single action, this approach reduces to LSTD (Least-Squares Temporal Differences) [5], that can be arbitrarily bad in an off-policy setting [20].

[3]A remarkable aspect of policy search is that it does not necessarily rely on the Markovian assumption, but this is out of the scope of this paper (residual approaches rely on it, through the Bellman equation). Some recent and effective approaches build on policy search, such as deep deterministic policy gradient [15] or trust region policy optimization [23]. Here, we focus on the canonical mean value maximization approach.

[4]Approximate linear programming could be considered as such but is often computationally intractable [8, 6].

[5]This choice is done for ease and clarity of exposition, the following results could be extended to continuous state and action spaces.

[6]The property $T_* v \geq v$ does not hold if $v$ is not the value function of a given policy, as in value-based approaches.

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
