[Supplementary Material]

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

*Proof.* The considered objective function can be rewritten as

$$\mathcal{J}_\nu(\pi) = \sum_{s \in \mathcal{S}} \nu(s) \left( \max_{a \in \mathcal{A}} q_\pi(s,a) - v_\pi(s) \right).$$

The classic policy gradient theorem [24] states that

$$\nabla(\nu v_\pi) = \sum_{s \in \mathcal{S}} \nu(s) \nabla v_\pi(s)$$

$$= \frac{1}{1-\gamma} \sum_{s \in \mathcal{S}} d_{\nu,\pi}(s) \sum_{a \in \mathcal{A}} \nabla \pi(a|s) q_\pi(s,a).$$

On the other hand, from basic subgradient calculus rules, we have that $\nabla \max_{a \in \mathcal{A}} q_\pi(s,a) = \nabla q_\pi(s, a_s^*)$, with $a_s^* \in \operatorname{argmax}_{a \in \mathcal{A}} q_\pi(s,a)$. Therefore:

$$\nabla \sum_{s \in \mathcal{S}} \nu(s) \max_{a \in \mathcal{A}} q_\pi(s,a)$$

$$= \sum_{s \in \mathcal{S}} \nu(s) \nabla q_\pi(s, a_s^*),$$

$$= \sum_{s \in \mathcal{S}} \nu(s) \nabla \left( \mathcal{R}(s, a_s^*) + \gamma \sum_{s' \in \mathcal{S}} P(s'|s, a_s^*) v_\pi(s') \right)$$

$$= \gamma \sum_{s \in \mathcal{S}} \nu(s) \sum_{s' \in \mathcal{S}} P(s'|s, a_s^*) \nabla v_\pi(s').$$

By noticing that $\nu(s) \sum_{s' \in \mathcal{S}} P(s'|s, a_s^*) = [\nu P_{\mathcal{G}(v_\pi)}](s)$ and that $\nabla \pi(a|s) = \pi(a|s) \nabla \ln \pi(a|s)$, we obtain the stated result. $\square$

# B Proof of Theorem 3

**Theorem 3** (Proxy bound for residual policy search). *We have that*

$$\|v_* - v_\pi\|_{1,\mu} \leq \frac{1}{1-\gamma} \left\| \frac{d_{\mu,\pi_*}}{\nu} \right\|_\infty \mathcal{J}_\nu(\pi) = \frac{1}{1-\gamma} \left\| \frac{d_{\mu,\pi_*}}{\nu} \right\|_\infty \|T_* v_\pi - v_\pi\|_{1,\nu}.$$

*Proof.* The proof can be easily derived from the analyses of [12], [17] or [22]. We detail it for completeness in the following.

First, notice that for any policy $\pi$ we have that $v_* \geq v_\pi$, thus $\|v_* - v_\pi\|_{1,\mu} = \mu(v_* - v_\pi)$. Using the fact that $v_\pi = (I - \gamma P_\pi)^{-1} \mathcal{R}_\pi$, we have:

$$v_{\pi'} - v_\pi = (I - \gamma P_{\pi'})^{-1} \mathcal{R}_{\pi'} - v_\pi$$

$$= (I - \gamma P_{\pi'})^{-1} (\mathcal{R}_{\pi'} + \gamma P_{\pi'} - v_\pi)$$

$$= (I - \gamma P_{\pi'})^{-1} (T_{\pi'} v_\pi - v_\pi).$$

Using this and the fact that $T_* v_\pi \geq T_{\pi'} v_\pi$, we have that

$$\mu(v_{\pi'} - v_\pi) = \mu(I - \gamma P_{\pi'})^{-1}(T_{\pi'} v_\pi - v_\pi)$$

$$= \frac{1}{1 - \gamma} d_{\mu,\pi'}(T_{\pi'} v_\pi - v_\pi)$$

$$\leq \frac{1}{1 - \gamma} d_{\mu,\pi'}(T_* v_\pi - v_\pi).$$

By the definition of the concentrability coefficient, $d_{\mu,\pi'} \leq \nu \| \frac{d_{\mu,\pi'}}{\nu} \|_\infty$, and as we assumed that $\nu(T_* v_\pi - v_\pi) \leq e$, we have

$$\mu(v_{\pi'} - v_\pi) \leq \frac{1}{1 - \gamma} \left\| \frac{d_{\mu,\pi'}}{\nu} \right\|_\infty \nu(T_* v_\pi - v_\pi)$$

$$\leq \frac{1}{1 - \gamma} \left\| \frac{d_{\mu,\pi'}}{\nu} \right\|_\infty e.$$

Choosing $\pi' = \pi_*$ gives the stated bound. $\qquad\square$

## C  More on the relation between the residual and the error

a. For RPS($\mu$).                b. For PS($\mu$).

Figure 4: Error as a function of the residual.

As said at the end of the experimental section, one could argue that the way we optimize the considered objective function is rather naive (for example, considering a constant learning rate). But this does not change the conclusions of this experimental study, that deals with how the error and the Bellman residual are related and with how the concentrability influences each optimization approach.

To expand on this, we consider the experiment of Sec. 4.1, where the distribution of interest is directly used. As this distribution is uniform, the concentrability coefficient is bounded (by the number of states), whatever the MDP is. Recall that for this experiment, we generated 100 Garnets and ran both algorithms for 1000 iterations, measuring the normalized error and the Bellman residual.

In Fig. 4, we show the error as a function of the Bellman residual for these experiments. These are the same data that where used for Fig. 1, presented in a different manner. Each curve corresponds to the learning in one MDP. Fig. 4.a shows the error as a function of the Bellman residual for RPS($\mu$), and Fig. 4.b shows the same thing for PS($\mu$).

The important thing is that these figures depend weakly on how the optimization is performed. A wise choice of the meta-parameters (or even of the optimization algorithm) will influence how fast and how well the objective criterion is optimized, but not on the mapping from residual to errors (or the converse).

Fig. 4.a shows this link for RPS. To see how learning processes, take the start of a curve in the upper-right of the graph (high residual) and follow it up to the left (low residual). As the residual decreases, so the error does (depending also on the concentrability of the MDP), which is consistent with the bound of Th. 3.

Fig. 4.b shows this link for PS. To see how learning processes, take the start of a curve in the upper-left of the graph (high error), and follow it up to the bottom (low error). We do not observe the same behavior as before. This was to be expected, and shows mainly that the error is not a proxy to the residual. More importantly, it shows that for decreasing the error, it might be efficient to highly increase the residual. This suggests that the residual is a bad proxy to policy optimization, and that maximizing directly the mean value is much more efficient (and insensitive to the concentrability, according to the rest of the experiments, which is a very important point).