[Reviews · NeurIPS 2017]

Reviewer 1



The paper investigates a fundamental question in RL, namely whether we should directly maximize the mean value, which is the primary objective, or if it is suitable to minimize the Bellman residual, as is done by several algorithms. The paper provides new theoretical analysis, in the form of a bound on residual policy search. This is complemented by a series of empirical results with random (Garnet-type) MDPs. The paper has a simple take-home message, which is that minimizing the Bellman residual is more susceptible to mismatch between the sample distribution and the optimal distribution, compared to policy search methods that directly optimize the value. The papes is well written; it outlines a specific question, and provides both theoretical and empirical support for answering this question. The new theoretical analysis extends previous known results. It is a modest extension, but potentially useful, and easy to grasp. The empirical results are informative, and constructed to shed light on the core question. I would be interested in seeing analogous results for standard RL benchmarks, to verify that the results carry over to "real" MDPs, not just random, but that is a minor point. One weakness of the work might be its scope: it is focused on a simple setting of policy search, without tackling implication of the work for more complex settings more commonly used, such as TRPO, DDPG.

Reviewer 2



The paper discusses a very interesting topic: which measure (BR minimization and mean value maximization) is better for policy search? 1. The reviewer is interested in one possible reason besides the discussion in the paper: the Markov assumption. As we know, the Bellman residual assumption is more restrictive to the Markov assumption of the task, whereas the mean value is not so restrictive to that. In real tasks for policy search, many tasks may break the Markov assumption. Does this also account for the reason why mean-value is a better metric in policy search? 2. There is a missing reference for (projected) Bellman minimization search: Toward Off-Policy Learning Control with Function Approximation, by H Maei et.al. (using projected TD for off-policy control instead of just policy evaluation) There is also one question´╝Ü H Maei's method suffers from the latent learning problem: the optimal policy, though learned, is not manifest in behavior. Until finishing the learning process, the learned policy is not allowed to be expressed and used. The reviewer wonders if such problem (latent learning) exists in the policy-based BR minimization approach. Please explain why if it does not exist.

Reviewer 3



The paper sheds light on the question whether policy search by directly maximizing the mean value or as a proxy minimizing the Bellman residual is more effective. Leaving aside implementation and estimation issues, the effectiveness of both objectives is compared theoretically and empirically. The empirical study suggests that maximizing the mean value is superior as its performance does not deteriorate when the concentrability coefficient becomes high. Overall, the study presented in this paper is well executed and the paper is well written. While the results are certainly not exciting, the authors raise the valid point that both objectives are prevalent in RL and therefore such a study is of interest. GIven that the study reveals a mismatch between the theoretical and empirical performance of mean value maximization, I would have liked to see a discussion of whether performance bounds without a dependency on the concentrability coefficient are possible and what the challenges to prove such a bound are. Are there lower bounds available that have a dependency on related quantities? The font size in the plots is too small. Axis labels and ticks are not readable at all.